# Therapeutic Advances in the Treatment of Gastroesophageal Cancers

**DOI:** 10.3390/biom13050796

**Published:** 2023-05-06

**Authors:** Jenny J. Li, Jane E. Rogers, Kohei Yamashita, Rebecca E. Waters, Mariela Blum Murphy, Jaffer A. Ajani

**Affiliations:** 1Department of Gastrointestinal Medical Oncology, University of Texas M. D. Anderson Cancer Center, 1515 Holcombe Blvd, Houston, TX 77030, USA; jjli2@mdanderson.org (J.J.L.);; 2Department of Pharmacy Clinical Program, University of Texas M. D. Anderson Cancer Center, 1515 Holcombe Blvd, Houston, TX 77030, USA; 3Department of Gastroenterological Surgery, Graduate School of Medical Science, Kumamoto University, 1-1-1 Honjo, Kumamoto 860-8556, Japan; 4Department of Pathology, Division of Pathology/Lab Medicine, University of Texas M. D. Anderson Cancer Center, 1515 Holcombe Blvd, Houston, TX 77030, USA

**Keywords:** esophageal cancer, gastric cancer, immunotherapy, targeted therapy

## Abstract

Gastroesophageal cancers are a group of aggressive malignancies that are inherently heterogeneous with poor prognosis. Esophageal squamous cell carcinoma, esophageal adenocarcinoma, gastroesophageal junction adenocarcinoma, and gastric adenocarcinoma all have distinct underlying molecular biology, which can impact available targets and treatment response. Multimodality therapy is needed in the localized setting and treatment decisions require multidisciplinary discussions. Systemic therapies for treatment of advanced/metastatic disease should be biomarker-driven, when appropriate. Current FDA approved treatments include HER2-targeted therapy, immunotherapy, and chemotherapy. However, novel therapeutic targets are under development and future treatments will be personalized based on molecular profiling. Herein, we review the current treatment approaches and discuss promising advances in targeted therapies for gastroesophageal cancers.

## 1. Introduction

Gastroesophageal cancers, including cancers of the esophagus, gastroesophageal junction (GEJ), and stomach, represent some of the most common cancers worldwide, with an estimated 1.7 million new cases per year. When combined, they are the third leading cause of cancer-related deaths globally [1]. Esophageal cancers are comprised of two major histologic subtypes: squamous cell carcinoma (ESCC) and adenocarcinoma (EAC). While ESCC is the most common subtype worldwide, EAC is more common in the Western countries [2]. ESCC is mainly found in the upper to mid esophagus and is associated with tobacco and alcohol use, whereas EAC is mostly found in the distal esophagus and is associated with obesity, gastroesophageal reflux, and Barrett’s esophagus [3,4]. In addition to differences in risk factors and locations, these two subtypes also have distinct molecular biology and responses to treatment, which should lead to different treatment strategies [5]. In contrast, GEJ and gastric cancers are almost all adenocarcinomas (ACs). Gastric adenocarcinomas (GACs) can be further classified into intestinal and diffuse histologic subtypes based on Lauren classification [6]. The Cancer Genome Atlas Research Network has also defined four molecular subtypes of GAC (Epstein-Barr Virus [EBV]-positive, microsatellite instability [MSI], genomically stable, and chromosomal instability), which may have implications for future therapeutic development [7].

At diagnosis, only 18% of esophageal cancers and 28% of gastric cancers are localized [8,9]. Around 40% of newly diagnosed gastroesophageal cancers have distant metastatic spread. The 5-year overall survival (OS) is 20.6% (esophageal) and 33.3% (gastric) for all stages combined, but for those with distant metastatic disease, the 5-year OS is only around 5% [8,9]. The general treatment approach for locally advanced disease involves multimodality therapy. For those patients with unresectable or metastatic disease, treatment is with palliative systemic therapies. Recent advancements in the field have led to biomarker guided treatment options such as immunotherapy (IO) and HER-2 targeted therapy. These newer therapies have not only improved outcomes for advanced/metastatic disease, they are now also being investigated in earlier stage settings. Many exciting potential new molecular targets are also being explored in ongoing clinical trials, leading to a future of personalized therapies. In this review, we summarize the current treatment landscape of gastroesophageal cancers and discuss the promising emerging therapeutics.

## 2. Localized Disease

Primary tumor location determines management in localized ESCC, EAC, GEJ AC, and GAC [10,11]. Surgical resection remains the main curative approach. Neoadjuvant, adjuvant, and perioperative approaches with chemotherapy, immunotherapy, and radiation are used or being investigated to improve surgical outcomes. These approaches differ based on primary tumor location, pathology, and institutional (and geographical) preferences. In general, GEJ/EAC patients are designated by the Siewert classification. Siewert Type I is adenocarcinoma of the lower esophagus with the epicenter located within 1 cm to 5 cm above the anatomic GEJ, Siewert Type II is true carcinoma of the cardia with the tumor epicenter within 1 cm above and 2 cm below the GEJ, and Siewert Type III is subcardial carcinoma with the tumor epicenter between 2 cm and 5 cm below the GEJ, which infiltrates the GEJ and lower esophagus from below [11]. Types I and II are treated as esophageal cancer, while Type III is treated as gastric cancer. 

Localized ESCC and EAC are managed in a similar fashion currently, but in the future, diverse approaches are likely to emerge. Preoperative chemoradiation (CRT) is the most common approach. The Chemoradiotherapy for Oesophageal Cancer Followed by Surgery Study (CROSS) trial conducted by Van Hagen et al. established this approach as the standard of care [12]. CROSS was a phase 3 randomized trial with resectable EAC, ESCC, and GEJ AC patients. Selected patients were randomized to receive surgery alone (n = 188) or CRT with carboplatin plus paclitaxel followed by surgery (n = 178). Outcome improvements were seen in the CRT + surgery group, including more R0 (microscopically margin-negative) resections (92% vs. 69%, *p* < 0.001), improved median OS (49.4 months vs. 24 months, *p* = 0.003), and improved disease-free survival (not reached vs. 24.2 months, *p* < 0.001). Histological differences were seen. ESCCs had higher pathological complete responses (49%) compared to EACs (23%), *p* = 0.008. Improvements in OS also differed based on histology (ESCC median OS 81.6 months with CRT + surgery; EAC median OS 43.2 months). With both histologies showing improvement, CROSS established a standard approach for localized EAC and ESCC patients. Ten-year follow-up showed continued survival benefit over surgery alone; however, the results also showed that further improvements in this area are needed as the 5-year cure rate was <40% [13]. The outcome differences based on histology suggest that different pathways should be explored as treatment targets for ESCC compared to EAC.

Recently, immunotherapy via an anti-programmed death-1 (PD-1) antibody, nivolumab, has been incorporated into the treatment of localized esophageal and GEJ cancer patients. Following neoadjuvant CRT and surgery, those who have residual pathological disease in the surgical specimen are recommended adjuvant nivolumab for 1 year. This approach was established by the CheckMate 577 trial [14]. CheckMate 577 was a phase 3 randomized placebo-controlled trial of resected (R0) ESCC, EAC, and GEJ AC patients who underwent neoadjuvant CRT and had residual pathological disease in the primary, lymph node(s), or both. Patients were randomized to nivolumab for 1 year (n = 532) vs. placebo for 1 year (n = 262). Median disease-free survival was improved (22.4 months vs. 11 months, *p* < 0.001) in the adjuvant nivolumab group. 

Adjunctive treatment for GAC surgical patients continues to be an area lacking a standard approach. The Medical Research Council Adjuvant Gastric Infusional Chemotherapy (MAGIC) trial was a landmark trial that implemented a perioperative approach [15]. This phase 3 trial compared epirubicin + cisplatin + 5-FU (ECF) × 3 cycles preoperative and ECF × 3 cycles postoperative (n = 250) to surgery alone (n = 253). Five-year OS was improved with the perioperative arm (36%) compared to the surgery alone arm (23%), *p* = 0.009. However, only 42% of patients completed all 6 cycles of perioperative treatment. The FLOT4 trial was a phase 2/3 trial that improved upon the MAGIC regimen by comparing the MAGIC approach (n = 360) to 4 cycles of 5-FU + docetaxel + oxaliplatin (FLOT) preoperative and 4 cycles of FLOT postoperative (n = 356) [16]. Median OS was improved with the FLOT arm compared to the MAGIC arm (50 months vs. 35 months, *p* = 0.012). Again, the completion rate of all perioperative treatment was low (37% MAGIC arm and 46% FLOT arm completed all cycles).

In contrast to the perioperative approach, the ACTS-GC, JACCRO-GC07, CLASSIC, CRITICS, ARTIST, and MacDonald trials evaluated the role of adjuvant chemotherapy and chemoradiation. ACTS-GC was a Japanese trial that was the first to examine the role of adjuvant chemotherapy [17]. In this randomized phase 3 trial, patients with stage II or III GAC were randomized to receive S-1 (an oral fluoropyrimidine) (n = 529) or nothing (n = 530) following D2 gastrectomy (gastrectomy with extensive lymph node removal from stations 1 to 12a). The 3-year OS rate was 80.1% in the S-1 group and 70.1% in the surgery-only group (hazard ratio 0.68, *p* = 0.003). The JACCRO GC-07 trial, another randomized phase 3 trial of patients with stage II or III GACs, showed that the addition of docetaxel to S-1 as adjuvant therapy (n = 454) improved 3-year relapse-free survival (66% vs. 50%, hazard ratio 0.632, *p* < 0.001) compared to S-1 alone (n = 459) [18]. CLASSIC was another phase 3 trial that compared capecitabine + oxaliplatin × 6 months following D2 gastrectomy (n = 520) vs. gastrectomy alone (n = 515) [19]. Five-year disease-free survival was improved with the adjuvant chemotherapy approach (68% vs. 53%, *p* < 0.0001). The ARTIST and ARTIST-II trials evaluated the role of adjuvant chemoradiation, but both showed that the addition of radiation to adjuvant chemotherapy did not significantly reduce the rate of recurrence after D2 gastrectomy [20,21]. The CRITICS trial, a randomized phase 3 trial, evaluated perioperative chemotherapy (n = 393) vs. preoperative chemotherapy + postoperative chemoradiation (n = 395) [22]. However, postoperative chemoradiation did not improve OS compared to postoperative chemotherapy (37 months vs. 43 months, hazard ratio 1.01, *p* = 0.90). Despite these results, adjuvant chemoradiation may still have a role in patients who underwent D0 or D1 gastrectomy (gastrectomy with less extensive lymph node removal; D1 = complete removal of stations 1–6 nodes, D0 = any lymph node removal less than D1). The MacDonald trial assessed surgery + adjuvant chemoradiation (n = 281) vs. surgery alone (n = 275) [23]. The median OS was significantly longer in the chemoradiation group compared to the surgery only group (36 months vs. 27 months, *p* = 0.005, hazard ratio 1.52). Of note, 90% of the patients on this trial underwent D0 or D1 gastrectomy instead of D2 gastrectomy. There are now additional ongoing trials to evaluate the role of preoperative chemoradiation (TOPGEAR, NCT01924819), as well as preoperative chemotherapy vs. chemotherapy + chemoradiation vs. chemoradiation (CRITICS II, NCT02931890).

Because the CROSS, MAGIC, and FLOT trials all included patients with distal esophageal/GEJ AC, it is unclear whether these patients should be treated with neoadjuvant chemoradiation or perioperative chemotherapy. Neo-AEGIS, a phase 3 open label randomized controlled trial, is currently underway to compare the CROSS regimen vs. the MAGIC/FLOT regimen [24]. Preliminary results showed that the three-year OS were similar between both approaches (57% CROSS vs. 55% perioperative, hazard ratio 1.03). However, all other outcomes favored the CROSS approach (i.e., nodal downstaging, R0 re-section, pathological complete response, and major pathologic complete response). Of note, FLOT has now replaced MAGIC as the preferred perioperative regimen, but only 15% of the perioperative arm received FLOT. The ESOPEC trial, which compares CROSS only to FLOT in localized EAC patients, will perhaps yield more relevant data, and we await these results [25].

The current National Comprehensive Cancer Network (NCCN) guidelines for the treatment of localized gastroesophageal cancer patients include multiple strategies discussed above, including preoperative chemoradiation, perioperative chemotherapy, postoperative chemotherapy, and postoperative chemoradiation [10,11]. Management is decided at an institutional level. Future treatments will likely expand upon the use of immune checkpoint inhibitors in the neoadjuvant setting (KEYNOTE-975, Rationale-311, CRISEC) [26,27,28]. IO use is especially promising in the microsatellite instability-high/deficient mismatch repair (MSI-H/dMMR) population (INFINITY, NEONIPIGA) [29,30]. We look forward to identifying more customized approaches based on molecular subtypes for the treatment of localized gastroesophageal cancer patients.

## 3. Metastatic Disease

Current standard of care treatment for metastatic gastroesophageal cancers involves systemic therapies such as chemotherapy, immunotherapy, and targeted therapies. Treatment is not curative, but the goal of treatment is to prolong survival, reduce cancer-related symptoms, and improve quality of life. Frontline therapy is guided by biomarkers, including HER2, microsatellite/mismatch repair status, and programmed-death-ligand-1 (PD-L1) expression (Table 1). Subsequent therapies are less well defined and treatment decision will depend on multiple factors including prior therapies, histology, biomarkers/mutations, and patient performance status.

### 3.1. First Line Systemic Therapies

For decades, chemotherapy with a fluoropyrimidine (5-fluorouracil or capecitabine) combined with a platinum (oxaliplatin or cisplatin) had been the standard treatment for all metastatic gastroesophageal cancers regardless of location or histology [31,32]. However, this approach results in a median OS of less than a year, so many attempts have been made to improve outcomes.

The ToGA trial established HER2 as a biomarker in gastroesophageal ACs for selecting HER2-directed therapy. This phase 3 randomized controlled trial evaluated the efficacy of trastuzumab, a monoclonal antibody against HER2, in combination with chemotherapy for first line treatment of HER2-positive gastric or GEJ AC [33]. The addition of trastuzumab to chemotherapy improved median OS compared to chemotherapy alone (13.8 months vs. 11.1 months, hazard ratio 0.74, *p* = 0.0046). An updated OS analysis 1 year after the final analysis showed median OS of 13.1 months vs. 11.7 months, hazard ratio 0.80 [34]. More recently, the KEYNOTE-811 trial further improved HER2-targeted regimens by adding the anti-PD-1 antibody pembrolizumab to trastuzumab and chemotherapy in patients with HER2-positive gastric or GEJ AC [35]. Initial findings from this phase 3 randomized placebo-controlled trial showed that the combination of chemotherapy, trastuzumab, and pembrolizumab significantly increased objective response rate (ORR) when compared to chemotherapy and trastuzumab (74.4% vs. 51.9%, *p* = 0.00006). Complete responses were also more frequent in the pembrolizumab group (11.3% vs. 3.1%). These results led to the FDA approval for the addition of pembrolizumab to chemotherapy and trastuzumab for first line treatment of patients with HER2-positive gastric and GEJ AC, and this regimen has become the new standard of care.

For patients with HER2-negative disease, treatment is guided by PD-L1 status. CheckMate 649, a phase 3 open-label randomized trial, evaluated the use of nivolumab + chemotherapy (n = 789) vs. chemotherapy alone (n = 792) in first line treatment of HER2-negative advanced EAC, GEJ AC, and GAC patients [36]. The median OS was significantly longer for the nivolumab + chemotherapy group when compared to the chemotherapy alone group in patients with a PD-L1 combined positive score (CPS) of five or greater (14.4 months vs. 11.1 months, hazard ratio 0.71, *p* < 0.0001). The hazard ratio for OS in patients with CPS < 5 was 0.94 (0.78–1.13). The 3-year follow up data continued to demonstrate longer survival for the nivolumab + chemotherapy group [37]. CheckMate 648, a phase 3 open-label randomized trial of nivolumab combination therapy in advanced ESCC, also showed benefit of IO in first line treatment of advanced disease [38]. Both the nivolumab + chemotherapy group (n = 321) and the nivolumab + ipilimumab group (n = 325) achieved significantly longer median OS than the chemotherapy alone group (n = 324) in patients with tumor cell PD-L1 expression of 1 or greater (15.4 months vs. 13.7 months vs. 9.1 months, respectively; *p* < 0.001 for nivolumab + chemotherapy vs. chemotherapy, *p* = 0.001 for nivolumab + ipilimumab vs. chemotherapy). Data from 29 -month follow up also continued to show survival benefit [39]. The KEYNOTE-590 trial evaluated the use of another IO, pembrolizumab, in combination with chemotherapy for first line treatment of advanced HER2-negative esophageal and GEJ ACs or ESCCs [40]. This phase 3 randomized controlled trial enrolled patients to receive either pembrolizumab + chemotherapy (n = 373) or placebo + chemotherapy (n = 376). Patients with a CPS of 10 or greater had significantly longer OS with pembrolizumab + chemotherapy vs. placebo + chemotherapy (13.5 months vs. 9.4 months, hazard ratio 0.62, *p* < 0.0001). Based on these trials, the American Society of Clinical Oncology (ASCO) treatment guidelines for HER2-negative advanced gastroesophageal cancers are as follows: GAC/EAC/GEJ AC with PD-L1 CPS ≥ 5, nivolumab + chemotherapy is recommended; EAC/GEJ AC/ESCC with PD-L1 CPS ≥ 10, pembrolizumab + chemotherapy is recommended; ESCC with PD-L1 tumor proportion score (TPS) ≥ 1, nivolumab + chemotherapy or nivolumab + ipilimumab are recommended [41].

MSI-H/dMMR represents a rare subgroup of gastroesophageal cancer patients. Landmark trials of PD-1 blockade with pembrolizumab in MSI-H/dMMR tumors (KEYNOTE-016, KEYNOTE-164, KEYNOTE-012, KEYNOTE-028, KEYNOTE-158) led to the tumor agnostic FDA approval of pembrolizumab use in the second line setting. Post hoc analysis of MSI-H/dMMR gastric or GEJ AC patients from the KEYNOTE-059, KEYNOTE-061, and KEYNOTE-062 trials showed that MSI-H/dMMR may be a predictive biomarker for response to pembrolizumab regardless of line of therapy [42], and there is ongoing discussion on the use of PD-1 blockade in the first line.

Unfortunately, patients with triple negative gastroesophageal cancers (HER2-negative, PD-L1 negative, MSS/pMMR) do not benefit from the addition of IO or HER2-targeted therapy to chemotherapy. Therefore, these patients should be treated with chemotherapy only (fluoropyrimidine + platinum). Trials are underway to identify additional biomarkers and targeted therapies to improve outcomes for this difficult to treat patient population.

### 3.2. Second Line and beyond Systemic Therapies

The recommended treatment for patients with advanced HER2-positive gastric or GEJ AC who have progressed on prior trastuzumab-containing regimen is trastuzumab deruxtecan, an antibody-drug conjugate of trastuzumab linked to a topoisomerase I inhibitor. This recommendation is based on the results of the DESTINY-Gastric01 trial, which is a phase 2 open-label randomized trial of trastuzumab deruxtecan (n = 125) vs. physician’s choice of chemotherapy (n = 62; 55 received irinotecan, 7 received paclitaxel) in patients with HER2-positive gastric or GEJ AC after progressing on 2 prior lines of therapy [43]. ORR was significantly higher in the trastuzumab deruxtecan group compared to the chemotherapy group (51% vs. 14%, *p* < 0.001), as was the OS (12.5 months vs. 8.4 months, hazard ratio 0.59, *p* = 0.01). These results led to the FDA approval of trastuzumab deruxtecan as a treatment option for advanced HER2-positive gastric or GEJ AC in the second line or later setting. The DESTINY-Gastric04 trial is currently underway to evaluate trastuzumab deruxtecan compared to ramucirumab + paclitaxel, which is the current standard chemotherapy regimen for second line treatment.

The role of IO therapy in the second line and beyond was evaluated by the KEYNOTE-181 and ATTRACTION-3 trials. KEYNOTE-181 was a phase 3 randomized controlled trial of pembrolizumab (n = 314) vs. investigator’s choice of chemotherapy (n = 314; docetaxel, paclitaxel, or irinotecan) in esophageal cancer patients after 1 prior line of standard therapy [44]. Pembrolizumab significantly improved median OS compared to chemotherapy in patients with PD-L1 CPS ≥ 10 (9.3 months vs. 6.7 months, hazard ratio 0.69, *p* = 0.0074), and the subgroup of patients with the highest benefit was ESCC with PD-L1 CPS ≥ 10 (OS 10.3 months vs. 6.7 months, hazard ratio 0.64). ATTRACTION-3, a phase 3 open-label randomized trial in patients with advanced ESCC after receiving 1 prior line of chemotherapy, assessed nivolumab (n = 210) vs. investigator’s choice of chemotherapy (n = 209; paclitaxel or docetaxel) [45]. Nivolumab significantly improved median OS compared to chemotherapy regardless of PD-L1 expression level. Thus, both pembrolizumab (PD-L1 CPS ≥ 10) and nivolumab (PD-L1 CPS agnostic) are IO options for second-line therapy in ESCC patients.

Chemotherapy remains the standard of care treatment for patients who are not eligible to receive trastuzumab deruxtecan or IO therapy (such as those at high-risk for lung toxicity or triple negative patients). The RAINBOW trial established ramucirumab + paclitaxel as the preferred regimen for second line treatment of advanced gastric or GEJ AC patients. This was a phase 3 randomized placebo-controlled trial that randomized patients to receive ramucirumab + paclitaxel (n = 330) vs. placebo + paclitaxel (n = 335) [46]. The ramucirumab + paclitaxel group had significantly longer OS than the control group (9.6 months vs. 7.4 months, hazard ratio 0.807, *p* = 0.017). Recently, results from the RAMIRIS trial suggested that FOLFIRI (5-fluorouracil, irinotecan) + ramucirumab may be an alternative treatment option in patients with previous exposure to docetaxel (such as those who received FLOT) and neurotoxicity [47]. This phase 2 randomized study compared FOLFIRI + ramucirumab (n = 72) vs. paclitaxel + ramucirumab (n = 38) in patients with advanced gastric or GEJ AC who progressed on first line chemotherapy. While this trial did not meet its primary end point of 6-month OS rate ≥ 65% in the FOLFIRI + ramucirumab arm, patients with prior exposure to docetaxel had more objective responses (25% vs. 8%) as well as longer progression free survival (PFS) (hazard ratio 0.49) in the FOLFIRI arm than the paclitaxel arm, and OS was similar between the two arms (hazard ratio 0.81). Other chemotherapy options for subsequent lines of therapy include single agent taxanes (docetaxel, paclitaxel), irinotecan, ramucirumab, trifluridine + tipiracil, and regorafenib [48,49,50,51,52,53].

### 3.3. Other Targeted Therapies

In addition to gastroesophageal cancer-specific approvals, the FDA has also granted approval for several targeted agents in a tumor agnostic manner based on specific molecular aberrations. Examples include entrectinib and larotrectinib for NTRK gene fusion-positive tumors, dabrafenib and trametinib for BRAF V600E mutated tumors, selpercatinib for RET gene fusion-positive tumors, and pembrolizumab for tumor mutation burden (TMB)-high tumors [54,55,56,57,58]. Even though these genetic alterations are rare in gastroesophageal cancers, treatment with these targeted agents can be considered in select patients with the appropriate alterations who have progressed on all available standard therapies. Ever expanding agnostic options suggest that we should be recommending next generation sequencing (NGS) testing when tumor tissue is available. The role of liquid biopsy is emerging.

## 4. Emerging Therapeutics

Over the years, many attempts have been made to explore new therapeutic targets in gastroesophageal cancers. Disappointingly, many of the prior trials of targeted therapies, whether used alone or in combination with chemotherapy, have been negative. Examples include trastuzumab emtansine (HER2), lapatinib (HER2), cetuximab (EGFR), panitumumab (EGFR), bevacizumab (VEGF), onartuzumab (MET), rilotumumab (MET), everolimus (mTOR), and olaparib (PARP) [59,60,61,62,63,64,65,66,67,68]. Despite this, recent efforts have brought to attention several promising new targets that may lead to new standard of care therapies (Table 2).

### 4.1. Claudin 18.2

Claudin 18 isoform 2 (CLDN 18.2) is a tight junction molecule that is orthotopically expressed in gastric cancers and ectopically activated in multiple other cancer types, including pancreatic, esophageal, ovarian, and lung [69]. Zolbetuximab, a monoclonal antibody against CLDN 18.2, has previously shown activity when added to chemotherapy in the phase 2 FAST trial, where the addition of zolbetuximab to chemotherapy increased survival of patients with advanced gastric/GEJ AC [70]. During the 2023 ASCO Gastrointestinal Cancer Symposium, results from the phase 3 SPOTLIGHT trial were presented. In HER2-negative, CLDN 18.2-positive, advanced gastric or GEJ AC patients, first line treatment with zolbetuximab + FOLFOX (5-fluorouracil + oxaliplatin) (n = 283) significantly improved PFS (10.6 months vs. 8.7 months, hazard ratio 0.75, *p* = 0.0066) and OS (18.2 months vs. 15.5 months, hazard ratio 0.75, *p* = 0.0053) compared to placebo + FOLFOX (n = 282) [71]. GLOW, another phase 3 trial of zolbetuximab + chemotherapy (capecitabine + oxaliplatin) in CLDN 18.2-positive HER2-negative advanced gastric/GEJ AC patients, recently announced that it met its the primary endpoint of PFS and secondary end point of OS [72]. These positive trials could establish CLDN 18.2 as a new predictive biomarker and zolbetuximab + chemotherapy as a new standard of care treatment for CLDN18.2-positive advanced gastric/GEJ cancer patients. It is important to note that in the SPOTLIGHT trial, only 13.2% patients who were CLDN 18.2-positive also had PD-L1 CPS ≥ 5, and all enrolled patients were HER2-negative. The CLDN 18.2-positive population represents a unique subset of patients who were previously mostly considered triple negative and would not have qualified for any targeted therapy options.

There are now multiple ongoing trials of novel monoclonal antibodies, antibody-drug conjugates, bispecific antibodies, and cellular therapies targeting CLDN 18.2 (NCT04856150, NCT04400383, NCT04632108, NCT04805307, NCT04495296, NCT04404595). Zolbetuximab is also being studied in combination with IO agents (NCT03505320).

### 4.2. FGFR2

Fibroblast growth factor receptors (FGFRs) have been found to be therapeutic targets in multiple cancer types, and FGFR2 specifically is a promising target for gastroesophageal cancers [73,74]. Bemarituzumab is a monoclonal antibody that targets FGFR2b [75]. The FIGHT trial, a phase 2 randomized placebo-controlled study, evaluated bemarituzumab + FOLFOX (n = 77) vs. placebo + FOLFOX (n = 78) in patients with FGFR2b overexpressed/amplified advanced gastric or GEJ AC [76]. While the trial did not meet its primary endpoint of PFS, the bemarituzumab group trended toward longer PFS (9.5 months vs. 7.4 months, hazard ratio 0.68, *p* = 0.073), longer OS (not reached vs. 12.9 months, hazard ratio 0.58, *p* = 0.027), and higher ORR (47% vs. 33%) compared to the placebo group. The FIGHT trial also showed that around 30% patients with HER2-negative advanced gastric cancer had FGFR2b overexpression or FGFR2 amplification, which supports the development of FGFR2 as a therapeutic target. The phase 3 FORTITUDE-101 trial is currently underway to further evaluate bemarituzumab + FOLFOX as first line treatment of FGFR2b-selected advanced gastric/GEJ AC patients (NCT05052801).

### 4.3. DKK1

The canonical Wnt/β-catenin signaling pathway plays an important role in cancer cell proliferation, survival, and migration [77,78,79]. Dickkopf-1 protein (DKK1) is an antagonist of the Wnt signaling pathway, and overexpression of DKK1 has been associated with tumor growth, angiogenesis, and a more immunosuppressive tumor microenvironment [80,81,82]. DKN-01 is a monoclonal antibody against DKK1, and it is being studied in combination with tislelizumab (anti-PD-1) and chemotherapy (FOLFOX or CAPOX) in the phase 2 DisTinGuish trial (NCT04363801). Data from this combination as first line therapy for HER2-negative advanced gastric/GEJ cancer patients (n = 25) showed promising activity [83,84]. The ORR for all patients is 68% and the ORR for the DKK1 high group is 90%. We await further data on this treatment combination.

### 4.4. Tyrosine Kinase Inhibitors

The combination of multi-kinase inhibitor (MKI) and IO has been successful in the treatment of other tumor types, including endometrial cancer and renal cell carcinoma. This combination is also being actively explored in gastroesophageal cancers. Prior studies have shown that oncogenic kinases have immunomodulatory activity and can lead to an immunosuppressive tumor microenvironment [85]. Inhibition of oncogenic kinases with MKIs have been shown to increase cytotoxic T-cell infiltration and decrease tumor associated macrophages, which leads to a more immune-permissive tumor microenvironment [86]. The impact of MKIs on the tumor microenvironment can synergize with IO use to improve antitumor efficacy [87]. The EPOC1706 trial was a single-arm phase 2 trial that evaluated lenvatinib + pembrolizumab in patients with advanced GAC (n = 29) and ORR was 69% [88]. There are now phase 3 trials to evaluate pembrolizumab + lenvatinib + chemotherapy in advanced ESCC and gastroesophageal AC (LEAP-014, LEAP-015). Similarly, trials to evaluate other combinations of MKIs and IOs are also underway (atezolizumab + cabozantinib, tislelizumab + sitravatinib; NCT05007613, NCT05461794).

### 4.5. Others

In addition to the MSI-H group, the EBV-positive molecular subgroup of GAC patients may present a unique opportunity for treatment with IO therapy. A prospective phase 2 trial of pembrolizumab in patients with advanced gastric or GEJ AC who progressed after first line chemotherapy found that EBV-positive patients (n = 6) had an ORR of 100% [89]. Another prospective observational study noted that EBV-positivity is a biomarker for immunotherapy in metastatic gastric cancer [90]. Due to the small sample sizes, larger prospective trials will be needed to fully evaluate the role of IO in EBV-positive patients. Nonetheless, targeting EBV-positive patients with IO therapy may be a novel option for the management of these patients.

## 5. Conclusions

In conclusion, the management of gastroesophageal cancers remains challenging. Locoregional disease require a multidisciplinary approach and may incorporate chemotherapy, radiation, and surgery. Treatment of advanced/metastatic disease is still palliative. Recent advancements in HER2-targeted therapies and immunotherapy bring a biomarker-driven approach to improve patient outcomes. Many trials are underway to improve therapies for current targets (i.e., evaluating the use of IO in locoregional disease, newer HER2-targeted agents such as margetuximab and zanidatamab), and even more trials are assessing new molecular targets based on histologies to usher in an era of personalized therapies. Monoclonal antibodies, bispecific T-cell engagers, antibody-drug conjugates, vaccines, and cellular therapies are all treatment modalities under exploration. We will need to increase our understanding of the genomic, transcriptomic, and proteomic landscape of gastroesophageal cancers to develop better future trials.

## Figures and Tables

**Table 1 biomolecules-13-00796-t001:** Major trials of immunotherapy and targeted therapy in advanced gastroesophageal cancers.

Trial	Treatment Regimen	Treatment Setting	Disease Site	Histologic Subtype	Biomarker
KEYNOTE-811	Chemotherapy + trastuzumab + pembrolizumab	1L	Gastric, GEJ	AC	HER2 positive, PD-L1 agnostic
CheckMate 649	Chemotherapy + nivolumab	1L	Esophageal, Gastric, GEJ	AC	HER2 negative, PD-L1 CPS ≥ 5
KEYNOTE-590	Chemotherapy + pembrolizumab	1L	Esophageal, GEJ	AC, SCC	HER2 negative, PD-L1 CPS ≥ 10
CheckMate 648	Chemotherapy + nivolumab, ipilimumab + nivolumab	1L	Esophageal	SCC	HER2 negative, PD-L1 TPS ≥ 1
DESTINY-Gastric01	Trastuzumab deruxtecan	2L+	Gastric, GEJ	AC	HER2 positive
KEYNOTE-181	Pembrolizumab	2L+	Esophageal	SCC	HER2 negative, PD-L1 CPS ≥ 10
ATTRACTION-3	Nivolumab	2L+	Esophageal	SCC	HER2 negative, PD-L1 agnostic

Abbreviations: 1L, first-line; 2L+, second-line and later; GEJ, gastroesophageal junction; AC, adenocarcinoma; SCC, squamous cell carcinoma.

**Table 2 biomolecules-13-00796-t002:** Select recently completed or ongoing trials of targeted therapy in gastroesophageal cancer.

Study Title	Phase	Intervention	Treatment Setting	Conditions	NCT Identifier	Biomarker/Target
A Phase 3 Efficacy, Safety and Tolerability Study of Zolbetuximab (Experimental Drug) Plus mFOLFOX6 Chemotherapy Compared to Placebo Plus mFOLFOX6 as Treatment for Gastric and Gastroesophageal Junction (GEJ) Cancer (**SPOTLIGHT**)	3	FOLFOX + Zolbetuximab	Metastatic/Advanced 1L	Gastric AC, GEJ AC	NCT03504397	Claudin 18.2
A Study of Zolbetuximab (IMAB362) Plus CAPOX Compared With Placebo Plus CAPOX as First-line Treatment of Subjects With Claudin (CLDN) 18.2-Positive, HER2-Negative, Locally Advanced Unresectable or Metastatic Gastric or Gastroesophageal Junction (GEJ) Adenocarcinoma (**GLOW**)	3	CAPOX + Zolbetuximab	Metastatic/Advanced 1L	Gastric AC, GEJ AC	NCT03653507	Claudin 18.2
A Study to Assess the Antitumor Activity, Safety, Pharmacokinetics and Biomarkers of Zolbetuximab (IMAB362) in Participants With Claudin (CLDN) 18.2 Positive, Metastatic or Advanced Unresectable Gastric and Gastroesophageal Junction (GEJ) Adenocarcinoma (**ILUSTRO**)	2	Zolbetuximab, Zolbetuximab + FOLFOX, Zolbetuximab + pembrolizumab, Zolbetuximab + FOLFOX ± nivolumab	Metastatic/Advanced 1L/3L+	Gastric AC, GEJ AC	NCT03505320	Claudin 18.2
A Study of Bemarituzumab (FPA144) Combined With Modified FOLFOX6 (mFOLFOX6) in Gastric/Gastroesophageal Junction Cancer (**FIGHT**)	2	FOLFOX + Bemarituzumab	Metastatic/Advanced 1L	Gastric AC, GEJ AC	NCT03694522	FGFR2b
Bemarituzumab or Placebo Plus Chemotherapy in Gastric Cancers With Fibroblast Growth Factor Receptor 2b (FGFR2b) Overexpression (**FORTITUDE-101**)	3	FOLFOX + Bemarituzumab	Metastatic/Advanced 1L	Gastric AC, GEJ AC	NCT05052801	FGFR2b
A Study of DKN-01 in Combination With Tislelizumab ± Chemotherapy in Patients With Gastric or Gastroesophageal Cancer (**DisTinGuish**)	2	FOLFOX/CAPOX + tislelizumab + DKN-01	Metastatic/Advanced 1L/2L	Gastric AC, GEJ AC	NCT04363801	DKK1
Efficacy and Safety of Pembrolizumab (MK-3475) Plus Lenvatinib (E7080/MK-7902) Plus Chemotherapy in Participants With Metastatic Esophageal Carcinoma (MK-7902-014/E7080-G000-320/**LEAP-014**)	3	Pembrolizumab + lenvatinib + chemotherapy	Metastatic/Advanced 1L	Esophageal SCC	NCT04949256	TKI
Efficacy and Safety of Lenvatinib (E7080/MK-7902) Plus Pembrolizumab (MK-3475) Plus Chemotherapy in Participants With Advanced/Metastatic Gastroesophageal Adenocarcinoma (MK-7902-015/E7080-G000-321/**LEAP-015**)	3	Pembrolizumab + lenvatinib + chemotherapy	Metastatic/Advanced 1L	Gastroesophageal AC	NCT04662710	TKI
Second-line Cabozantinib and Atezolizumab in Patients With Recurrent or Metastatic Esophageal Squamous Cell Carcinoma	2	Atezolizumab + cabozantinib	Metastatic/Advanced 2L	Esophageal SCC	NCT05007613	TKI
Study To Investigate the Efficacy and Safety of Sitravatinib in Combination With Tislelizumab in Participants With Esophageal Squamous Cell Carcinoma	2	Tislelizumab + sitravatinib	Metastatic/Advanced 2L+	Esophageal SCC	NCT05461794	TKI

Abbreviations: 1L, first-line; 2L, second-line; 3L, third-line; GEJ, gastroesophageal junction; AC, adenocarcinoma; SCC, squamous cell carcinoma.

## Data Availability

Not applicable.

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
