# Peer review of "Therapeutic Advances in the Treatment of Gastroesophageal Cancers"

_biomolecules, 2023, doi:10.3390/biom13050796_

Round 1

Reviewer 1 Report

The review article provides a comprehensive and insightful overview of the current state of research and treatment options for gastroesophageal cancers. The authors highlight the aggressive nature and poor prognosis of these malignancies, and emphasize the heterogeneity of each subtype, with distinct underlying molecular biology that impacts treatment decisions and available targets.

The review emphasizes the importance of multimodality therapy in the localized setting, and the need for multidisciplinary discussions to optimize treatment outcomes. The authors also stress the importance of biomarker-driven systemic therapies for advanced/metastatic disease, when appropriate, and highlight the current FDA-approved treatments including HER2-targeted therapy, immunotherapy, and chemotherapy.

In addition, the review provides insights into novel therapeutic targets under development and emphasizes the need for personalized treatment approaches based on molecular profiling. The authors also discuss promising advances in targeted therapies for gastroesophageal cancers, which hold great promise for improving outcomes in patients with these malignancies.

Overall, the review article provides a clear and concise overview of the current state of research and treatment options for gastroesophageal cancers. The authors present a balanced perspective, highlighting both current treatment options and the potential for future advances in personalized treatment approaches. This review will be a valuable resource for researchers, clinicians, and patients alike.

I have several small comments to improve the manuscript:

1. Primary tumor location determines management in localized ESCC, EAC, GEJ AC, 58 and GAC. Surgical resection remains the only curative approach. Localized ESCC and EAC are managed in the same fashion currently.

ESCC can also be curatively treated with definitive CRT according to ESMO guidelines. Therefore, ESCC and EAC are NOT managed in the same fashion currently

Author Response

Response to Reviewer 1 Comments

Point 1:

Primary tumor location determines management in localized ESCC, EAC, GEJ AC, 58 and GAC. Surgical resection remains the only curative approach. Localized ESCC and EAC are managed in the same fashion currently.

ESCC can also be curatively treated with definitive CRT according to ESMO guidelines. Therefore, ESCC and EAC are NOT managed in the same fashion currently

Response 1: We appreciate the insightful comment by Review 1. Pratice patterns between Europe and United States may be sligthly different. At our institution, both ESCC and EAC patients may be treated curatively with definitive CRT if they are not surgical candidates, though the rates of pathologic complete response is much higher in ESCC than in EAC. We have updated the text to reflect that localized ESCC and EAC are managed similarly (not the same). We also updated that surgical resection is the main curative approach (not the only one).

Reviewer 2 Report

The article is definitely exhaustive. In my opinion, nothing has to be added.

Author Response

Response to Reviewer 2 Comments

Point 1:  

The article is definitely exhaustive. In my opinion, nothing has to be added.

Response 1: We appreciate Reviewer 2’s comments

Reviewer 3 Report

In my opinion, the authors of the article: “Therapeutic advances in the treatment of gastroesophageal cancers” undertook an interesting and simultaneously hard subject to study and discuss. This subject is very important especially for doctors from cancer clinics. Gastric cancer is still among leading cancers with generally poor prognosis and survival rates. Any works summarizing treatment possibilities can lead to better understanding of the problem. The authors carried out huge work to summarize different ways of gastroesophageal cancers treatment. The problem is serious due to no treatments which could lead to full recovery. The authors suggest that maybe, in the future, more personalized therapy would be possible (what would be especially interesting and promising).

I suggest that the manuscript could be accepted for publication after minor revision.

Specific comments:

Line 29 – citation [1] could taken because in the next statement there is the same reference;

Line 77 – R0-  abbreviation – what does it mean?

Lines 127, 132 – the abbreviations D0, D1, D2 – should be explained – maybe they are common, but not for all the readers.

Generally the English language is ok. The article is well written. Maybe, in some points it needs minor editing in English language.

Author Response

Response to Reviewer 3 Comments

Point 1:

Line 29 – citation [1] could taken because in the next statement there is the same reference;

Response 1: We have removed the citation.

Point 2:

Line 77 – R0-  abbreviation – what does it mean?

Response 2: We have added the definition for R0 resection

Point 3:

Lines 127, 132 – the abbreviations D0, D1, D2 – should be explained – maybe they are common, but not for all the readers.

Response 3: We have added the definitions for D0, D1, and D2 gastrectomty